# Early acquisition of S-specific Tfh clonotypes after SARS-CoV-2 vaccination is associated with the longevity of anti-S antibodies

**Xiuyuan Lu[1†], Hiroki Hayashi[2†], Eri Ishikawa[1,3†], Yukiko Takeuchi[1], Julian Vincent Tabora Dychiao[1], Hironori Nakagami[2*], Sho Yamasaki[1,3,4*]**

[1]Laboratory of Molecular Immunology, Immunology Frontier Research Center, Osaka University, Suita, Japan; [2]Department of Health Development and Medicine, Osaka University Graduate School of Medicine, Suita, Japan; [3]Department of Molecular Immunology, Research Institute for Microbial Diseases, Osaka University, Suita, Japan; [4]Center for Infectious Disease Education and Research (CiDER), Osaka University, Suita, Japan

**\*For correspondence:**
nakagami@gts.med.osaka-u.ac.jp (HN);
yamasaki@biken.osaka-u.ac.jp (SY)

[†]These authors contributed equally to this work

**Abstract** SARS-CoV-2 vaccines have been used worldwide to combat COVID-19 pandemic. To elucidate the factors that determine the longevity of spike (S)-specific antibodies, we traced the characteristics of S-specific T cell clonotypes together with their epitopes and anti-S antibody titers before and after BNT162b2 vaccination over time. T cell receptor (TCR) αβ sequences and mRNA expression of the S-responded T cells were investigated using single-cell TCR- and RNA-sequencing. Highly expanded 199 TCR clonotypes upon stimulation with S peptide pools were reconstituted into a reporter T cell line for the determination of epitopes and restricting HLAs. Among them, we could determine 78 S epitopes, most of which were conserved in variants of concern (VOCs). After the 2nd vaccination, T cell clonotypes highly responsive to recall S stimulation were polarized to follicular helper T (Tfh)-like cells in donors exhibiting sustained anti-S antibody titers (designated as 'sustainers'), but not in 'decliners'. Even before vaccination, S-reactive CD4+ T cell clonotypes did exist, most of which cross-reacted with environmental or symbiotic microbes. However, these clonotypes contracted after vaccination. Conversely, S-reactive clonotypes dominated after vaccination were undetectable in pre-vaccinated T cell pool, suggesting that highly responding S-reactive T cells were established by vaccination from rare clonotypes. These results suggest that de novo acquisition of memory Tfh-like cells upon vaccination may contribute to the longevity of anti-S antibody titers.

## eLife assessment

This **important** study by Lu et al aimed to determine the key factors of T cell responses associated with durable antibody responses following the initial two shots of COVID-19 mRNA vaccinations. By comparing the SARS-CoV-2 spike protein (S)-specific T cell subsets between 'Ab sustainers' and 'Ab decliners' that were present post-vaccination, the authors concluded that S-specific CD4+ T cells in 'Ab sustainers' were enriched with Tfh cells. There is **solid** evidence as the authors applied multiple methods and approaches to address the key questions, and the presented data are robust.

## Introduction

The pandemic COVID-19, caused by the severe acute respiratory syndrome coronavirus 2 (SARS-CoV-2), has expanded worldwide (*Hu et al., 2021*). Many types of vaccines have been developed or in basic and clinical phases to combat infection and deterioration of COVID-19 (*Creech et al., 2021*; *Krammer, 2020*). Among them, messenger ribonucleic acid (mRNA) vaccines, BNT162b2/Comirnaty and mRNA-1273/Spikevax, have been approved with over 90% efficacy at 2 months post-2nd dose vaccination (*Baden et al., 2021*; *Polack et al., 2020*), and widely used. Pathogen-specific antibodies are one of the most efficient components to prevent infection. Yet, mRNA vaccine-induced serum antibody titer is known to be waning over 6 months (*Levin et al., 2021*; *Pegu et al., 2021*). Accordingly, the effectiveness of the vaccines decreases over time, and thus multiple doses and repeated boosters are necessary (*Andrews et al., 2022*).

The production and sustainability of spike (S)-specific antibody could be related to multiple factors, especially in the case of humans (*Collier et al., 2021*; *Levin et al., 2021*). Among them, the characteristics of SARS-CoV-2-specific T cells are critically involved in the affinity and longevity of the antibodies (*Crotty, 2019*; *Nelson et al., 2022*; *Terahara et al., 2022*). Elucidation of the key factors of T cell responses that contribute to the durable immune responses induced by vaccination would provide valuable information for the vaccine development in the future. However, the relationship between antibody sustainability and the types of antigen-specific T cells has not been investigated in a clonotype resolution.

Recent studies reported that S-reactive T cells pre-existed before exposure to SARS-CoV-2 (*Grifoni et al., 2020*; *Le Bert et al., 2020*; *Mateus et al., 2020*; *Meckiff et al., 2020*; *Sekine et al., 2020*). Common cold human coronaviruses (HCoVs) including strains 229E, NL63, OC43, and HKU1 are considered major cross-reactive antigens that primed these pre-existing T cells (*Becerra-Artiles et al., 2022*; *Low et al., 2021*; *Loyal et al., 2021*; *Mateus et al., 2020*), while bacterial cross-reactive antigens were also reported (*Bartolo et al., 2022*; *Lu et al., 2021*). However, the functional relevance of cross-reactive T cells during infection or vaccination is still in debate.

In this study, both humoral and cellular immune responses were evaluated at 3, 6, and 24 weeks after BNT162b2/Comirnaty vaccination. S-specific T cells before and after vaccination were analyzed on clonotype level using single-cell-based T cell receptor (TCR) and RNA sequencing to determine their characteristics and epitopes in antibody sustainers and decliners. These analyses suggest the importance of early acquisition of S-specific Tfh cells in the longevity of antibodies.

## Results

### SARS-CoV-2 mRNA vaccine elicits transient humoral immunity

Blood samples were collected from a total of 43 individuals (*Table 1*) who had no SARS-CoV-2 infection history when they received two doses of SARS-CoV-2 mRNA vaccine BNT162b2. Samples were taken before and after the vaccination (*Figure 1A*). Consistent with the previous report (*Polack et al., 2020*), most participants exhibited more severe side effects after 2nd dose of vaccination than 1st

**Table 1.** Demographic data of the participants.

| | Percentage (number) |
|---|---|
| Total number | 100% (43) |
| Age group | |
| 20–39 | 39.5% (17) |
| 40–49 | 30.2% (13) |
| 50–59 | 25.6% (11) |
| 60–69 | 4.7% (2) |
| Sex | |
| Male | 60.5% (26) |
| Female | 39.5% (17) |

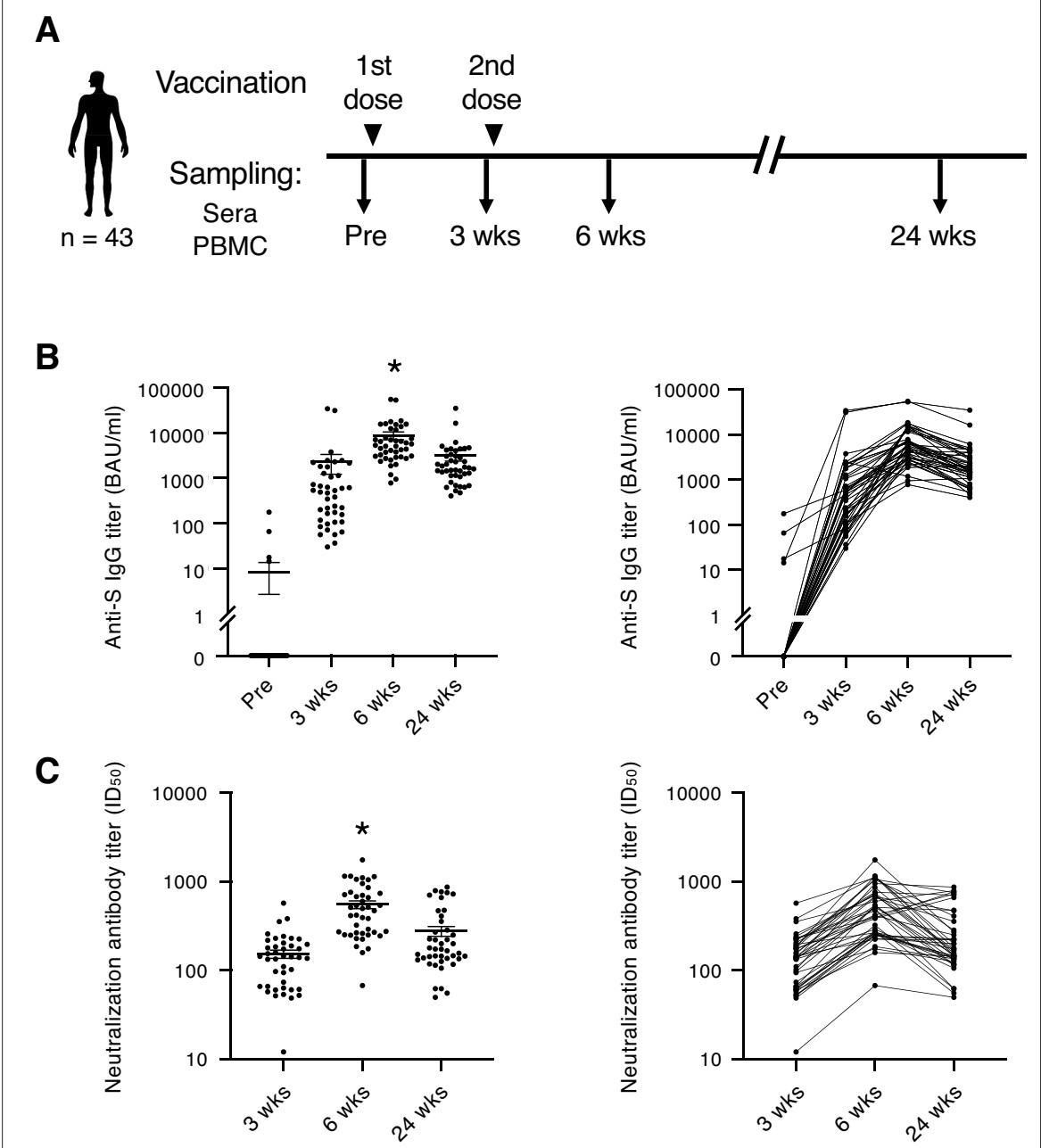

**Figure 1.** SARS-CoV-2 mRNA vaccine elicits transient humoral immunity. (**A**) Vaccination and sampling timeline of blood donors in this study. (**B**) Anti-S IgG titer of serum samples was determined by ELISA. Mean ± SEM (left) and individual data (right) are shown. *, p<0.05 vs. Pre, 3 weeks, 24 weeks, respectively. (**C**) Neutralization activity (ID50) of serum samples was determined by pseudo-virus assay. Mean ± SEM (left) and individual data (right) are shown. *, p<0.05 vs 3 weeks, 24 weeks, respectively. Wks, weeks.

The online version of this article includes the following figure supplement(s) for figure 1:

**Figure supplement 1.** Humoral immune response of BNT162b2 vaccinees.

dose locally (*Table 2*) and systemically (*Table 3*). At 3 weeks, anti-S IgG antibody titer increased in most participants. At 6 weeks, anti-S antibody titer was at its peak. S antibody titer gradually decreased over 24 weeks (*Figure 1B*). The antibody titer was reduced by 56.8% on average. Donors of different sexes or age groups showed no significant difference in anti-S antibody titer (*Figure 1— figure supplement 1*). The neutralization activity of the post-vaccinated sera showed similar tendency with the anti-S antibody titer during the study period (*Figure 1C*). The above results indicate that the

**Table 2.** Demographic data of the reported clinical adverse effects (at injection site).

| | Percentage (number) |
| --- | --- |
| Swelling (injection site) | |
| After 1st dose | 27.9% (12) |
| After 2nd dose | 51.2% (22) |
| Sore/pain (injection site) | |
| After 1st dose | 88.4% (38) |
| After 2nd dose | 86.0% (37) |
| Warmth (injection site) | |
| After 1st dose | 32.6% (14) |
| After 2nd dose | 41.9% (18) |

mRNA vaccine effectively activated humoral immune responses in healthy individuals, but decreased by 24 weeks over time as reported (*Levin et al., 2021*; *Pegu et al., 2021*).

## Antibody sustainers had highly expanded S-reactive Tfh clonotypes

To address the role of T cells in maintaining the antibody titer, we analyzed the S-responsive T cells in the post-vaccination samples from eight donors, among whom four donors showed relatively sustained anti-S antibody titer during 6 weeks to 24 weeks (reduction <30%; sustainers, donors #8, #25, #27, and #28), while the other four donors showed largely declined anti-S antibody titer (reduction >80%; decliners, donors #4, #13, #15, and #17; *Figure 2A* and *Figure 2—figure supplement 1A*). The possibility of SARS-CoV-2 infection of sustainers was ruled out by analyzing anti-nucleocapsid protein (N) antibody titer in the sera samples at 24 weeks (*Figure 2—figure supplement 1B*). Antibody sustainability did not correlate with bulk T cell responses to S protein, such as IFNγ production (*Figure 2—figure supplement 1C*).

To enrich the S-reactive T cells, we labeled the peripheral blood mononuclear cells (PBMCs) with a cell proliferation tracer and stimulated the PBMCs with an S peptide pool for 10 days. Proliferated T cells were sorted and analyzed by single-cell TCR- and RNA-sequencing (scTCR/RNA-seq). Clustering analysis was done with pooled samples of three time points from eight donors, and various T cell subtypes were identified (*Figure 2B*, *Source code 1*). We found that, overall, the S-reactive T cells did not skew to any particular T cell subset (*Figure 2B*). However, by grouping the cells from decliners and sustainers separately, we found difference in the frequency of the cells within the circled population (*Figure 2C*), and overall, the sustainer individuals had more cells in this region (*Figure 2—figure supplement 2*). These cells showed high Tfh signature scores and expressed characteristic genes of Tfh cells (*Figure 2D*). This tendency became more pronounced when we selected highly expanded (top 16) clonotypes in each donor (*Figure 2E*). In sustainers, S-specific Tfh clusters appeared from 6 weeks (*Figure 2F*), suggesting that vaccine-induced Tfh-like cells that have potency of deriving to Tfh cells were established immediately after 2nd vaccination.

## Identification of dominant S epitopes recognized by vaccine-induced T cell clonotypes

To elucidate the epitopes of the highly expanded clonotypes, we reconstituted their TCRs into a T cell hybridoma lacking endogenous TCRs and having an NFAT-GFP reporter gene. These cell lines were stimulated with S peptides using transformed autologous B cells as antigen-presenting cells (APCs). The epitopes of 53 out of 128 reconstituted clonotypes were successfully determined (*Figure 3*, *Table 4*, *Figure 3—figure supplements 1 and 2*). Epitopes of expanded Tfh cells were not limited in any particular region of S protein (*Figure 3*). About 72% of these epitopes conserved in Delta and Omicron variants (*Tables 4 and 5*). Within the rest of 28% of epitopes which were mutated in variants of concern (VOCs), although some mutated epitopes located in the receptor-binding domain (RBD) of VOCs lost antigenicity, recognition of most epitopes outside the RBD region was maintained or rather

**Table 3.** Demographic data of the reported clinical adverse effects (systemic symptoms).

| | Percentage (number) |
|---|---|
| **Fever** | |
| After 1st dose | |
| Mild (37.5 °C ≥) | 2.3% (1) |
| Severe (≥38.0 °C) | 0% (0) |
| After 2nd dose | |
| Mild (37.5 °C ≥) | 25.6% (11) |
| Severe (≥38.0 °C) | 23.3% (10) |
| **Fatigue** | |
| After 1st dose | |
| Mild | 18.6% (8) |
| Severe | 0% (0) |
| After 2nd dose | |
| Mild | 67.4% (29) |
| Severe | 18.6% (8) |
| **Headache** | |
| After 1st dose | |
| Mild | 7.0% (3) |
| Severe | 0% (0) |
| After 2nd dose | |
| Mild | 32.6% (14) |
| Severe | 7.0% (3) |
| **Chill** | |
| After 1st dose | |
| Mild | 4.7% (2) |
| Severe | 0% (0) |
| After 2nd dose | |
| Mild | 23.3% (10) |
| Severe | 9.3% (4) |
| **Nausea** | |
| After 1st dose | |
| Mild | 0% (0) |
| Severe | 0% (0) |
| After 2nd dose | |
| Mild | 4.7% (2) |
| Severe | 0% (0) |
| **Diarrhea** | |
| After 1st dose | |
| Mild | 0% (0) |
| Severe | 0% (0) |

*Table 3 continued on next page*

*Table 3 continued*

| | Percentage (number) |
|---|---|
| After 2nd dose | |
| Mild | 0% (0) |
| Severe | 0% (0) |
| Muscle pain | |
| After 1st dose | |
| Mild | 48.8% (21) |
| Severe | 0% (0) |
| After 2nd dose | |
| Mild | 55.8% (24) |
| Severe | 4.7% (2) |
| Joint pain | |
| After 1st dose | |
| Mild | 4.7% (2) |
| Severe | 0% (0) |
| After 2nd dose | |
| Mild | 25.6% (11) |
| Severe | 4.7% (2) |

increased in the variants (*Table 5* and *Figure 3—figure supplement 3*). These results suggest that the majority of S-reactive clonotypes after vaccination can respond to antibody-escaping VOCs.

## Identification of S epitopes and cross-reactive antigens of pre-existing T cell clonotypes

Before the pandemic, T cells cross-reacting to S antigen were present in the peripheral blood (*Grifoni et al., 2020*; *Le Bert et al., 2020*; *Mateus et al., 2020*; *Meckiff et al., 2020*; *Sekine et al., 2020*). To characterize these pre-existing S-reactive cells, we analyzed the PBMCs collected from donors who consented to blood sample donation before vaccination (#4, #8, #13, #15, and #17). PBMCs were stimulated with the S peptide pool for 10 days, and proliferated T cells were sorted and analyzed by scTCR/RNA-seq. Similar to vaccine-induced S-reactive T cells (*Figure 2B*), characteristics of pre-existing S-reactive T cells were diverse (*Figure 4A*, *Source code 1*). To track the dynamics of cross-reactive clones after vaccination, we combined the single-cell sequencing data of pre- and post-vaccinated PBMCs and analyzed the clonotypes that have more than 50 cells in total (*Figure 4B*). We did find some cross-reactive clonotypes that were further expanded by vaccination, and most of these clonotypes had cytotoxic features, being CD8[+] effector memory T cells (Tem) or minor CD4[+] cytotoxic T cells (CTLs). In contrast, most of the cross-reactive CD4[+] T cells became minor clonotypes after vaccination.

We also explored the epitopes of the top 16 expanded clonotypes in each pre-vaccinated donor by reconstituting the TCRs into reporter cell lines. We identified 18 epitopes from S protein and determined some possible cross-reactive antigens (*Figure 5*, *Table 6*, *Figure 5—figure supplement 1*). Most of these cross-reactive antigens originated from environmental or symbiotic microbes (*Table 6*). Furthermore, majority of the reactive T clonotypes showed regulatory T cell (Treg) signatures (*Figure 5*). Six of these 80 analyzed clonotypes could also be frequently detected in the public TCR database Adaptive (*Emerson et al., 2017*; *Nolan et al., 2020*). Notably, most of these clonotypes, except for one case, showed comparable frequencies between pre-pandemic healthy donors and COVID-19 patients (*Figure 6*), suggesting that these clonotypes did not expand upon SARS-CoV-2 infection, despite they were present before the pandemic. Thus, it is unlikely that these cross-reactive

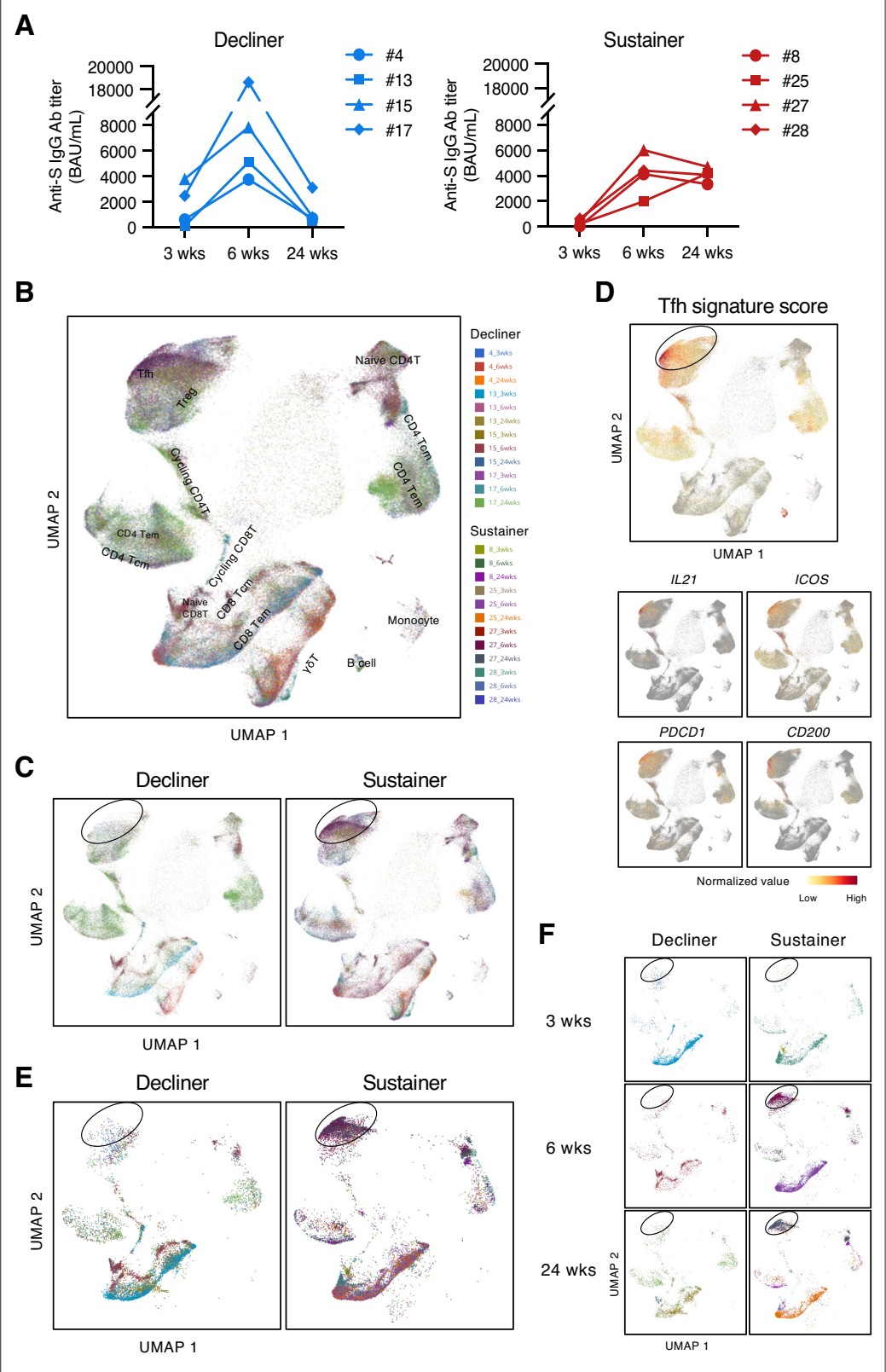

**Figure 2.** Antibody sustainers had highly expanded S-reactive Tfh clonotypes. (**A**) Anti-S IgG titer of serum samples from sustainers and decliners is shown individually. (**B, C, E, F**) UMAP projection of T cells in single-cell analysis of post-vaccinated samples collected from all donors. Each dot corresponds to a single cell and is colored according to the samples from different time points of donors. All samples together with annotated cell types

*Figure 2 continued on next page*

(**B**), samples grouped by donor type (decliners and sustainers) (**C**), top 16 expanded clonotypes (16 clonotypes that had the most cell numbers from each donor) grouped by donor type (**E**), and top 16 expanded clonotypes grouped by time point and donor type (**F**) are shown. Tcm, central memory T cells; Tem, effector memory T cells; Treg, regulatory T cells; γδT, γδ T cells. (**D**) Tfh signature score and expression levels of the canonical Tfh cell markers, *IL21*, *ICOS*, *PDCD1* and *CD200*, are shown as heat maps in the UMAP plot.

The online version of this article includes the following figure supplement(s) for figure 2:

**Figure supplement 1.** Humoral and cellular immune responses of sustainers and decliners.

**Figure supplement 2.** Sustainer individuals had more cells in the circled region than decliner individuals.

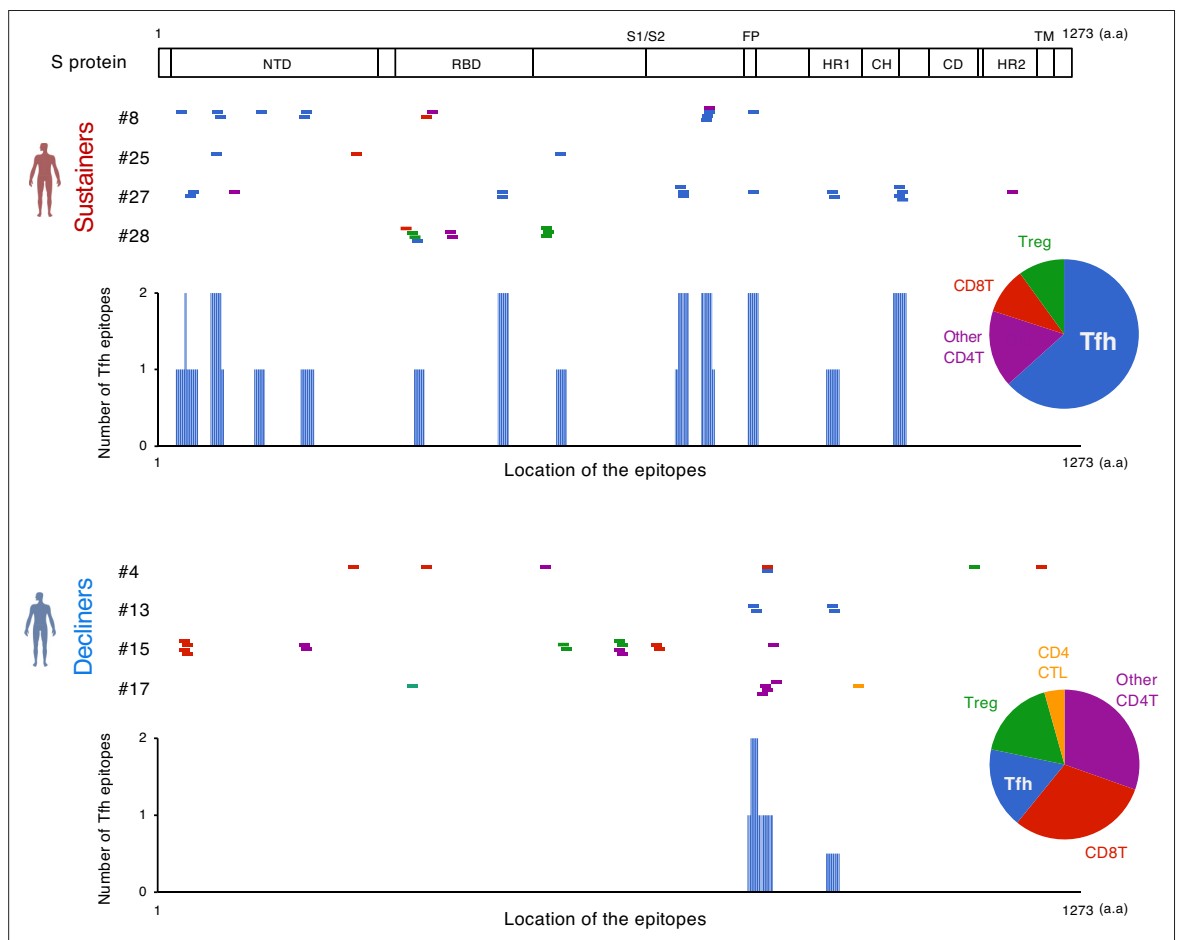

**Figure 3.** The location of S epitopes recognized by top expanded T clonotypes from post-vaccination samples. T cell S epitopes recognized by top expanded TCR clonotypes in post-vaccinated samples from sustainers and decliners are mapped by their locations in S protein. Each short bar indicates a 15-mer peptide that activated the TCRs. Epitopes are shown in different colors according to the subsets of the T cells they activated. Relative frequencies of the T cell subsets are shown in pie charts. Numbers of identified epitopes recognized by a dominant T subset in sustainers (Tfh) are shown in blue bars. NTD, N-terminal domain; RBD, receptor-binding domain; FP, fusion peptide; HR1, heptad repeat 1; CH, central helix; CD, connector domain; HR2, heptad repeat 2; TM, transmembrane domain.

The online version of this article includes the following figure supplement(s) for figure 3:

**Figure supplement 1.** Determination of S epitopes for post-vaccinated T cell clonotypes expanded in sustainers and decliners.

**Figure supplement 2.** Determination of restricting HLAs for post-vaccinated T cell clonotypes expanded in sustainers and decliners.

**Figure supplement 3.** Determination of mutated epitope antigenicity for post-vaccinated T cell clonotypes expanded in sustainers and decliners.

**Table 4.** TCR clonotypes expanded in post-vaccinated samples and their TCR usages, epitopes and restricting HLAs.

| Donor | Clonotype | TRBV | CDR3β | TRBJ | TRAV | CDR3α | TRAJ | S epitope* | Restricting HLA |
|---|---|---|---|---|---|---|---|---|---|
| | Post_4 | 11-2 | CASSPTGTNEKLFF | 1-4 | 13-1 | CAGGADGLTF | 45 | SFSTFKCVGVSPTKL$_{373-387}$† | DRA-DRB1*15:02 |
| | Post_5 | 19 | CASSGRPEGPQHF | 1-5 | 20 | CAVLNQAGTALIF | 15 | FKIYSKHTPIN$_{201-211}$ | DRA-DRB1*09:01 |
| | Post_6 | 11-2 | CASSLEGTEAFF | 1-1 | 5 | CAESRYMGRRALTF | 5 | FQFCNDPFLGVYYHK$_{131-147}$ | DPA1*01:03-DPB1*04:02 |
| | Post_7 | 2 | CAGLAGVDTGELFF | 2-2 | 5 | CAERVGRRALTF | 5 | YSVLYNSASFSTFKC$_{365-379}$ | A*24:02 |
| | Post_8 | 20-1 | CSATRDRRSYNEQFF | 2-1 | 12-2 | CAVLTNTGNQFYF | 49 | LLQYGSFCTQLNRAL$_{753-767}$ | DRA-DRB1*15:02 |
| | Post_9 | 7-9 | CASSLLGEQYF | 2-7 | 22 | CAGAGGTSYGKLTF | 52 | KRFDNPVLPFN$_{77-87}$ | DPA1*02:02-DPB1*05:01 |
| | Post_10 | 6-1 | CASSEGASNQPQHF | 1-5 | 12-1 | CVVNKGSSASKIIF | 3 | LLQYGSFCTQL$_{753-763}$ | DRA-DRB1*15:02 |
| | Post_12 | 20-1 | CSAYSIYNEQFF | 2-1 | 9-2 | CALSMNTGFQKLVF | 8 | PPAYTNSFTRGVYYP$_{25-39}$ | DRA-DRB1*09:01 |
| | Post_14 | 19 | CASRPNRGDNSPLHF | 1-6 | 12-1 | CVVSIGFGNVLHC | 35 | CSNLLLQYGSFCTQL$_{749-763}$ | DRA-DRB1*15:02 |
| #8 | Post_15 | 28 | CASSLMGGAYGYTF | 1-2 | 8-6 | CAVRRGGSGGSNYKLTF | 53 | SKRSFIEDLLFNKVT$_{813-827}$ | DPA1*01:03-DPB1*04:02 |
| | Post_7 | 7-9 | CAPSNANTGELFF | 2-2 | 12-1 | CVVNEADKLIF | 34 | YLQPRTFLLK$_{269-278}$ | A*02:01 |
| | Post_12 | 20-1 | CSARDVEVGSGYTF | 1-2 | 4 | CLVGPYNQGGKLIF | 23 | TGVLTESNKKFLPFQ$_{549-563}$ | DRA-DRB1*14:54 |
| #25 | Post_15 | 3-1 | CASSPLSGSSYEQYF | 2-7 | 12-1 | CVVGTDSWGKLQF | 24 | TNGTKRFDNPVLPFN$_{73-87}$ | DPA1*02:02-DPB1*05:01/DPA1*01:03-DPB1*05:01 |
| | Post_1 | 20-1 | CSAIAGDADTQYF | 2-3 | 9-2 | CALTSAAGNKLTF | 17 | NQFNSAIGKIQ$_{925-935}$ | DRA-DRB1*09:01 |
| | Post_2 | 30 | CAWNLGGGNQPQHF | 1-5 | 8-2 | CVVSERASSYKLIF | 12 | SKRSFIEDLLFNKVT$_{813-827}$ | DPA1*02:02-DPB1*04:02 |
| | Post_3 | 5-4 | CASSQGQGSYGYTF | 1-2 | 4 | CLVGDSDTGRRALTF | 5 | NFTISVTTEIL$_{717-727}$ | DRA-DRB1*09:01 |
| | Post_5 | 7-2 | CASGTGSYNEQFF | 2-1 | 12-2 | CAVKRGNQGGKLIF | 23 | STEIYQAGSTPCNGV$_{469-483}$ | DRA-DRB1*04:03 |
| | Post_7 | 6-6 | CASRLPGNRAQPQHF | 1-5 | 36/DV7 | CAVESGSSNTGKLIF | 37 | KSNIIRGWIFGTTLD$_{97-111}$ | DRA-DRB4*01:03 |
| | Post_8 | 6-5 | CASSYSGGTVTGELFF | 2-2 | 41 | CAVGIRGNEKLTF | 48 | KVFRSSVLHST$_{41-51}$ | DRA-DRB1*04:03 |
| | Post_9 | 20-1 | CSARDGQTATNEKLFF | 1-4 | 17 | CATNAGGTSYGKLTF | 52 | EIRASANLAAT$_{1017-1027}$ | DRA-DRB1*04:03 |
| | Post_11 | 30 | CAWSVKGFPSQHF | 1-5 | 6 | CALGSTSNTGKLIF | 37 | EIRASANLAAT$_{1017-1027}$ | DRA-DRB1*04:03 |
| | Post_13 | 5-6 | CASSSRTGYNSPLHF | 1-6 | 27 | CAGAKGSGTYKYIF | 40 | STEIYQAGSTPCNGV$_{469-483}$ | DRA-DRB1*04:03 |
| | Post_15 | 5-5 | CASSSDRNYGYTF | 1-2 | 12-1 | CVVNMVTGGYNKLIF | 4 | NFTISVTTEILPVSM$_{717-731}$ | DRA-DRB1*09:01 |
| #27 | Post_16 | 7-9 | CASSSQPGLAGVKIGNEQFF | 2-1 | 5 | CAEIPPSNTGKLIF | 37 | ISGINASVVNIQKEI$_{1169-1183}$ | DRA-DRB1*04:03 |
| | Post_5 | 3-1 | CASSQGGSEKLFF | 1-4 | 1-1 | CAVGGNTDKLIF | 34 | LVKNKCVNFNF$_{533-543}$ | DRA-DRB3*03:01 |
| | Post_10 | 12-3 | CASSSGRTGFGYTF | 1-2 | 30 | CGTEFGSEKLVF | 57 | VIRGDEVRQIA$_{401-411}$ | DRA-DRB3*03:01 |
| | Post_11 | 5-8 | CASSLQKTTGPSYGYTF | 1-2 | 8-6 | CAVSPYTGRRALTF | 5 | SVYAWNRKRIS$_{349-359}$ | DRA-DRB1*13:02 |
| | Post_12 | 18 | CASSASVDPTEAFF | 1-1 | 1-1 | CASFTGGGNKLTF | 10 | KSTNLVKNKCVNFNF$_{529-543}$ | DRA-DRB3*03:01 |
| | Post_14 | 7-6 | CASSLSGTGGTGELFF | 2-2 | 4 | CLVGDMRSGGGADGLTF | 45 | PFGEVFNATRFASVY$_{337-351}$ | B*40:01 |
| #28 | Post_15 | 6-2 | CASSYPPSGGRTGFGEAFF | 1-1 | 14/DV4 | CAMRDIGFGNVLHC | 35 | WNRKRISNCVADYSV$_{353-367}$ | DRA-DRB4*01:03 |

*Table 4 continued on next page*

*Table 4 continued*

| Donor | Clonotype | TRBV | CDR3β | TRBJ | TRAV | CDR3α | TRAJ | S epitope* | Restricting HLA |
|---|---|---|---|---|---|---|---|---|---|
| | Post_2 | 25-1 | CASTGDNYGYTF | 1-2 | 21 | CAINTGNQFYF | 49 | YVVGYLQPR$_{265-273}$ | A*33:03 |
| | Post_10 | 7-9 | CASRPSGTSREQYF | 2-7 | 29 | CAGNNAGNMLTF | 39 | FIKQYGDCLGDIAAR$_{833-847}$ | A*33:03 |
| | Post_11 | 7-9 | CASSTRTSGGGLSYEQYF | 2-7 | 3 | CAVNKAAGNKLTF | 17 | YSVLYNSASFSTFKC$_{365-379}$ | A*24:02 |
| | Post_13 | 20-1 | CASSIEQGDLGYTF | 1-2 | 23/DV6 | CAASIPNSGYALNF | 41 | FIKQYGDCLGDIAAR$_{833-847}$ | DQA1*01:02-DQB1*05:03 |
| | Post_14 | 5-6 | CASSPGQGILEQYF | 2-7 | 24 | CAFVPLSDGQKLLF | 16 | YIKWPWYIWL$_{1209-1218}$ | A*24:02 |
| | Post_15 | 7-3 | CASGIHTGELFF | 2-2 | 26-1 | CIVNNAGNMLTF | 39 | TDNTFVSGNCDVVIG$_{1117-1131}$ | DQA1*01:02-DQB1*06:04 |
| #4 | Post_16 | 7-6 | CASSPGPSEADTQYF | 2-3 | 1-1 | CAVRDGDDKIIF | 30 | KSTNLVKNKCVNFNF$_{529-543}$ | DRA-DRB3*03:01 |
| | Post_13 | 7-2 | CASSVGQSKGKSAETQYF | 2-5 | 22 | CAVNEYSGAGSYQLTF | 28 | SKRSFIEDLLFNKVT$_{813-827}$ | DPA1*01:03-DPB1*02:01 |
| | Post_15 | 20-1 | CSAGDTASTYGYTF | 1-2 | 9-2 | CALSDGAGNKLTF | 17 | NQFNSAIGKIQ$_{925-935}$ | DRA-DRB1*09:01 |
| #13 | Post_16 | 30 | CAWSLQGQRPQHF | 1-5 | 38-1 | CAFMKQRGGSEKLVF | 57 | FIEDLLFNKVTLADA$_{817-831}$ | DPA1*01:03-DPB1*02:01 |
| | Post_1 | 12-4 | CASSSHRDRGVEAFF | 1-1 | 12-1 | CVVNFDRGSTLGRLYF | 18 | TRGVYYPDKVF$_{33-43}$ | B*15:01 |
| | Post_6 | 3-1 | CASSQQLNTGELFF | 2-2 | 38-2/DV8 | CAYRKTSGTYKYIF | 40 | WRVYSTGSNVF$_{633-643}$ | DRA-DRB1*15:02 |
| | Post_7 | 28 | CASSFPDRYYSNQPQHF | 1-5 | 1-2 | CAVRAVGGNKLVF | 47 | TRGVYYPDKVF$_{33-43}$ | B*15:01 |
| | Post_9 | 27 | CASSPGHEQYF | 2-7 | 14/DV4 | CAMSPIRTYKYIF | 40 | RSVASQSIIAY$_{685-695}$ | B*15:01 |
| | Post_11 | 3-1 | CASSRELISEQYF | 2-7 | 38-2/DV8 | CAYKRTSGTYKYIF | 40 | WRVYSTGSNVF$_{633-643}$ | DRA-DRB1*15:02 |
| | Post_12 | 28 | CASSSYGTSGGRAEQFF | 2-1 | 16 | CALSGGLTGGGNKLTF | 10 | LGDIAARDLICAQKF$_{841-855}$ | DRA-DRB1*08:02 |
| | Post_13 | 30 | CAWRTGQGITSPLHF | 1-6 | 8-2 | CVVNNAGNMLTF | 39 | VFKNIDGYFKIYSKH$_{193-207}$ | DPA1*02:02-DPB1*05:01 |
| | Post_14 | 6-1 | CASSEAGGSGANVLTF | 2-6 | 9-2 | CALSGTGTYKYIF | 40 | KKFLPFQQFGR$_{557-567}$ | DPA1*02:02-DPB1*05:01 |
| #15 | Post_16 | 27 | CASSLGTINTGELFF | 2-2 | 17 | CATAPAGGTSYGKITF | 52 | IDGYFKIYSKHTPIN$_{197-211}$ | DRA-DRB1*08:02 |
| #15 | Post_4 | 6-2 | CASTSTARGSYNEQFF | 2-1 | 27 | CAGHSNTGNQFYF | 49 | TRFASVYAWNRKRIS$_{345-359}$ | DRA-DRB1*08:02 |
| | Post_10 | 9 | CASSKTSGAYNEQFF | 2-1 | 9-2 | CALDNARLMF | 31 | FIKQYGD$_{833-839}$ | DRA-DRB1*15:01 |
| #17 | Post_11 | 20-1 | CSARPPGGGNNEQFF | 2-1 | 26-2 | CILRDGTGANNLFF | 36 | QALNTLVKQLSSNFG$_{957-971}$ | DRA-DRB1*08:02 |
| | Post_15 | 7-9 | CASSLARGNSPLHF | 1-6 | 38-2/DV8 | CAFVGSQGNLIF | 42 | AARDLICAQKFNGLT$_{845-859}$ | DRA-DRB1*08:02 |

*Overlapped epitope sequence is shown when a clonotype recognized two or three sequential peptides.
[†]Number ranges indicate the location of peptides in the proteins.

**Table 5.** Reactivity of each clonotype to mutated epitopes in SARS-CoV-2 VOCs.

| Donor | Clonotype | Mutated epitopes in VOCs | | Domain | Response |
|---|---|---|---|---|---|
| #8 | Post_4 | Omicron BA.1<br>Omicron BA.2, 4/5 | PFFTFKCYGVSPTKL*<br>PFFAFKCYGVSPTKL | RBD | ↓<br>↓ |
| #8 | Post_5 | Omicron BA.1 | FKIYSKHTPII | non-RBD | ↑ |
| #8 | Post_6 | Delta, Omicron BA.2, 4/5<br>Omicron BA.1 | FQFCNDPFLDVYYHK<br>FQFCNDPFLD---HK | non-RBD | ↓<br>↓ |
| #8 | Post_7 | Omicron BA.1<br>Omicron BA.2, 4/5 | YSVLYNLAPFFTFKC<br>YSVLYNFAPFFAFKC | RBD | ↓<br>↓ |
| #8 | Post_8 | Omicron BA1, 2, 4/5 | LLQYGSFCTQLKRAL | non-RBD | ↑ |
| #8 | Post_10 | Omicron BA1, 2, 4/5 | LLQYGSFCTQLKRAL | non-RBD | ↑ |
| #27 | Post_5 | Delta<br>Omicron BA.1, 2, 4/5 | STEIYQAGSKPCNGV<br>STEIYQAGNKPCNGV | RBD | ↓<br>↓ |
| #27 | Post_13 | Delta<br>Omicron BA.1, 2, 4/5 | STEIYQAGSKPCNGV<br>STEIYQAGNKPCNGV | RBD | ↓<br>↓ |
| #28 | Post_5 | Omicron BA.1 | LVKNKCVNFNFNGLK | non-RBD | ↑ |
| #28 | Post_10 | Omicron BA.2, 4/5 | VIRGNEVSQIA | RBD | ↓ |
| #28 | Post_14 | Omicron BA.1, 2, 4/5 | PFDEVFNATRFASVY | RBD | ↓ |
| #4 | Post_11 | Omicron BA.1<br>Omicron BA.2, 4/5 | YSVLYNLAPFFTFKC<br>YSVLYNFAPFFAFKC | RBD | ↓<br>↓ |
| #15 | Post_9 | Delta<br>Omicron BA.1, 2, 4/5 | RRRARSVASQSIIAY<br>HRRARSVASQSIIAY | non-RBD | ↑<br>↑ |
| #15 | Post_16 | Omicron BA.1 | IDGYFKIYSKHTPII | non-RBD | → |
| #17 | Post_11 | Omicron BA.1, 2, 4/5 | QALNTLVKQLSSKFG | non-RBD | ↓ |
| #17 | Post_15 | Omicron BA.1 | AARDLICAQKFKGLT | non-RBD | ↓ |

*Amino acids colored red indicate mismatches compared with corresponding S epitopes of Wuhan strain.

T clonotypes contribute to the establishment of S-reactive T cell pools during either vaccination or infection.

## Discussion

Previous studies showed that Tfh function and germinal center development were impaired in deceased COVID-19 patients (*Kaneko et al., 2020*) and Tfh cell number correlated with neutralizing antibody (*Gong et al., 2020*; *Juno et al., 2020*; *Zhang et al., 2021*). Consistent with the above studies, we found that the donors having sustained antibody titers between 6 and 24 weeks post-vaccination had more S antigen-responsive Tfh-like clonotypes maintained in the periphery as a memory pool. As circulating Tfh clonotypes can reflect the population of germinal center Tfh cells (*Brenna et al., 2020*), it is possible that these maintained S-responsive Tfh cells contribute to the prolonged production of anti-S antibodies. These results imply that Tfh polarization of S-reactive T cells in the blood after 2nd vaccination can be a marker for the longevity of serum anti-S antibodies. Although monitoring of S-specific Tfh cells in germinal center is ideal (*Mudd et al., 2022*), it is currently difficult for outpatients in clinics.

Since the antigen used for BNT162b2 is a full-length S protein from the Wuhan-Hu-1 strain, it is important to estimate whether vaccine-induced Wuhan S-reactive T cells recognize neutralizing antibody-evading VOCs, such as Omicron variants. To investigate the dominant T cell epitopes of vaccine-developed T cells, we utilized a proliferation-based sorting strategy to enrich the S-responsive T cells. The limitation of this strategy is that a 10 day stimulation would change the transcriptional profile and repertoire of T cells. However, this strategy allowed us to select the T cell clonotypes that vigorously responded to the S antigen stimulation, while weakly responsive cells and anergic cells

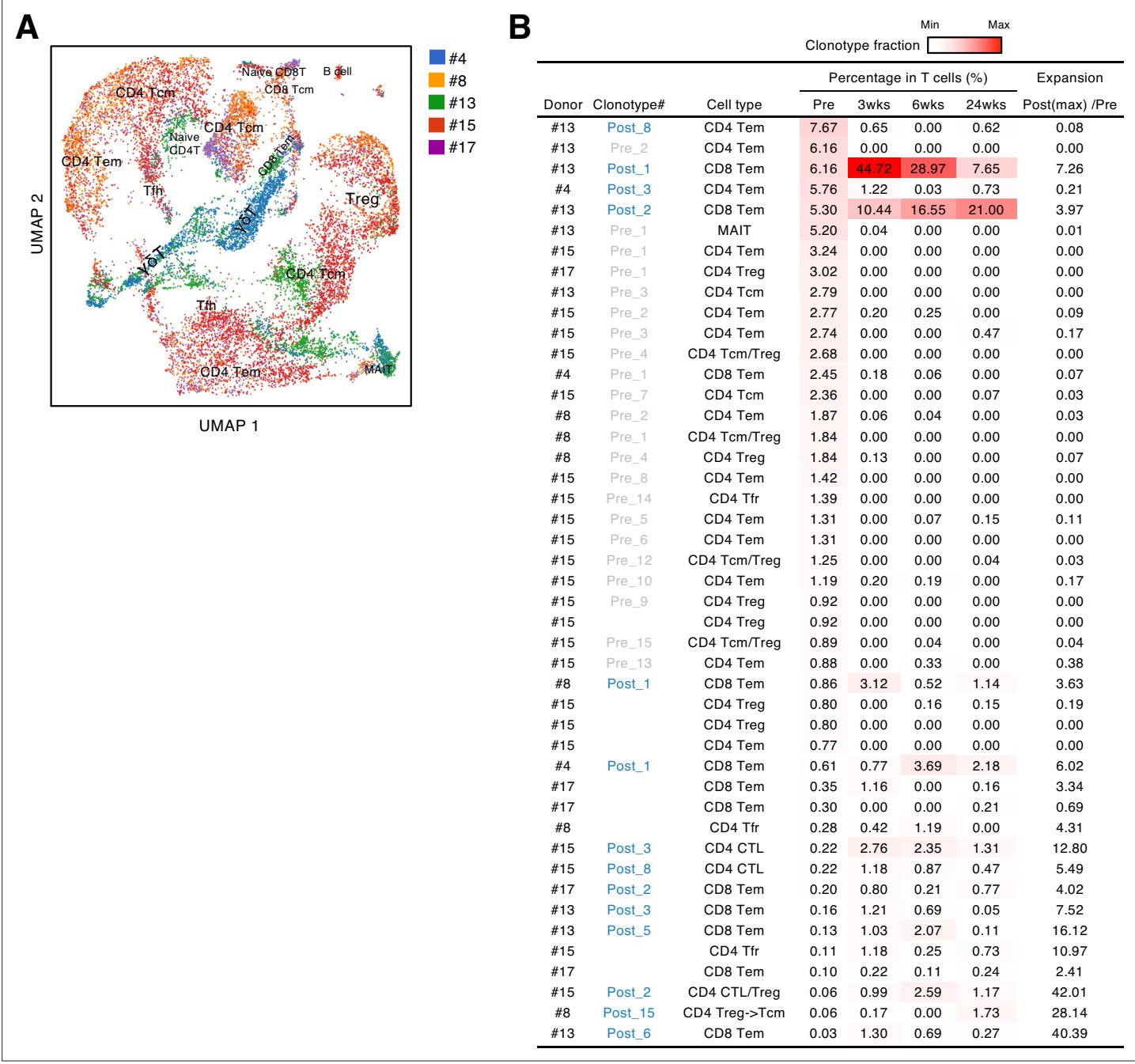

**Figure 4.** Characteristics and dynamics of S-cross-reactive clonotypes. (**A**) UMAP projection of T cells in single-cell analysis of pre-vaccinated samples from donors #4, #13, #15, #17, and #8. Each dot corresponds to a single cell and is colored according to the samples from different donors. Annotated cell types are shown. (**B**) Donor, name of reconstituted clonotypes, cell type, clonotype fraction in T cells from each time points, and expansion ratio of clonotypes that were found in pre-vaccinated samples and had more than 50 cells in the combined pre- and post-vaccinated sample set. For clonotypes that showed more than one type, the major type is listed in the front. The expansion ratio was calculated using the maximum cell fraction at post-vaccination points divided by the cell fraction at the pre-vaccination point of each clonotype. Clonotypes that have an expansion ratio larger than 1 are considered as expanded post-vaccination. Cell fractions at individual time points are shown as heat map. Tfr, follicular regulatory T cells; MAIT, mucosal-associated invariant T cells.

will be less considered, which is exactly in line with our purpose. Consistent with previous reports (*GeurtsvanKessel et al., 2022*; *Keeton et al., 2022*; *Tarke et al., 2022*), most of the epitopes determined in the current study were conserved in Delta and Omicron (BA.1, BA.2, and BA.4/5) strains, suggesting that vaccine-induced T cells are able to recognize the mutated S proteins from these

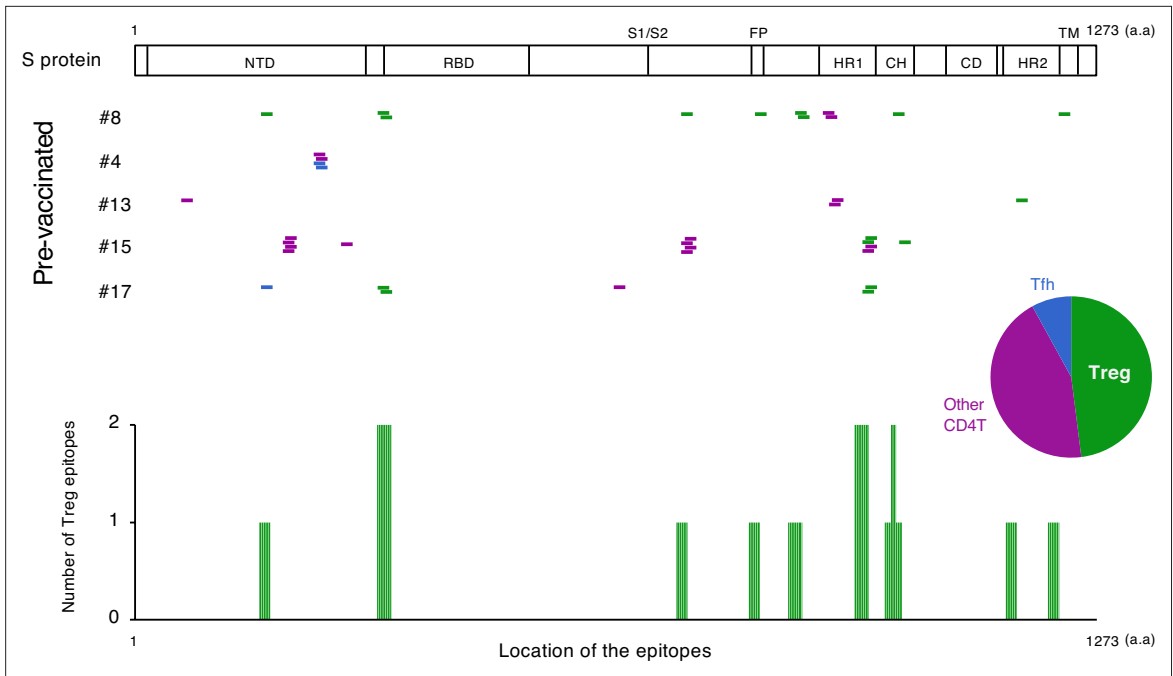

**Figure 5.** The location of S epitopes of pre-existing S-reactive T cells. S epitopes recognized by top expanded TCR clonotypes in pre-vaccinated samples are mapped by their locations in S protein. Each short bar indicates a 15-mer peptide that activated the TCRs. Epitopes are shown in different colors according to the subtypes of the T cells they activated. Relative frequencies of the T cell subtypes from all five donors are shown in the pie chart. Numbers of identified epitopes recognized by a dominant T subset of pre-existing clonotypes (Treg) from all donors are shown in green bars.

The online version of this article includes the following figure supplement(s) for figure 5:

**Figure supplement 1.** Determination of S epitopes, restricting HLAs and cross-reactive epitopes for pre-existing T cell clonotypes expanded by S stimulation.

**Figure supplement 2.** The pre-existing S-reactive T cell clonotypes did not recognize HCoV epitopes.

variants, despite the B epitopes being largely mutated in these VOCs (*GeurtsvanKessel et al., 2022*; *Tarke et al., 2022*).

SARS-CoV-2-recognizing T cells existed prior to exposure to the S antigens (*Grifoni et al., 2020*; *Le Bert et al., 2020*; *Mateus et al., 2020*; *Meckiff et al., 2020*; *Sekine et al., 2020*), which is consistent with our observation with PBMCs from donors who were uninfected and pre-vaccinated. Among these pre-existing S-reactive clonotypes, CD8[+] cytotoxic T clonotypes were expanded by the vaccination, whereas most CD4[+] T clonotypes became less dominant after vaccination (*Figure 4B*). Currently, the reason for the opposite tendency is unclear. In the present study, we showed that pre-existing T clonotypes cross-reacting to S protein are unlikely to contribute to vaccine-driven T cell immunity. This could be due to the fact that cross-reactive T cells had relatively low avidity to S protein (*Bacher et al., 2020*). Alternatively, but not mutually exclusively, considering that most of these cross-reactive T clonotypes have Treg signature (*Figure 5*), they could be developed to tolerate symbiotic or environmental antigens, and might be ineffective to the defense against SARS-CoV-2 and thus replaced by the other effective T clonotypes induced by vaccination. One exceptional pre-existing clonotype was #15-Pre_2, as they vigorously expanded in COVID-19 patients (*Figure 6*). This clonotype was clustered within a CD4[+] Tem population and cross-reactive to environmental bacteria, *Myxococcales bacterium* (*Table 6*). Thus, in some particular settings, clonotypes primed by common bacterial antigens might potentially contribute during infection.

Common cold human coronavirus (HCoV)-derived S proteins are reported as potential cross-reactive antigens for pre-existing SARS-CoV-2 S-reactive T cells (*Becerra-Artiles et al., 2022*; *Low et al., 2021*; *Loyal et al., 2021*; *Mateus et al., 2020*). However, the highly responding SARS-CoV-2 S-reactive clonotypes in pre-vaccinated donors did not react with HCoV S proteins in the present study (*Figure 5—figure supplement 2*), which might be partly due to the difference of cohorts or ethnicities. Instead, most of those T cells cross-reacted with environmental or symbiotic bacteria.

**Table 6.** S-cross-reactive TCR clonotypes expanded in pre-vaccinated samples and their TCR usages, epitopes, restricting HLAs and cross-reactive epitopes in microbes other than SARS-CoV-2.

| Donor | Clonotype | TRBV | CDR3β | TRBJ | TRAV | CDR3α | TRAJ | S epitope | Restricting HLA | Cross-reactive antigen [species] | Cross-reactive peptide | Post-vaccinated expansion |
|---|---|---|---|---|---|---|---|---|---|---|---|---|
| #4 | Pre_5 | 6-6 | CASSYPGGGSETQYF | 2-5 | 35 | CAGAVAVQGAQKLVF | 54 | LLALHRSY LTP$_{241-251}$ * | DRA-DRB1*14:54 | Phosphoribosyl formylglycinamidine cyclo-ligase [Firmicutes bacterium] | VAEALLAVHR SYLTP$_{220-234}$† | No |
| #4 | Pre_7 | 6-6 | CASSYPGSGGELFF | 2-2 | 21 | CAVENSGNTPLVF | 29 | LLALHRSY LTP$_{241-251}$ | DQA1*01:04-DQB1*05:03 | Phosphoribosylf ormylglycinamidine cyclo-ligase [Firmicutes bacterium] | VAEALLAVHR SYLTP$_{220-234}$ | No |
| #8 | Pre_1 | 6-2 | CASRPNRGRFRGNQPQHF | 1-5 | 23/DV6 | CAGEEKETSGSRLTF | 58 | NCTFEYVSQP FLMDL$_{165-179}$ | DRA-DRB1*15:02 | Fumarylacetoacetate hydrolase family protein [Alcaligenes faecalis] Hypothetical protein [Planctomycetales bacterium] | ASLIEVSQP FLLEP$_{225-239}$ AAGFEYVSQ PFSLP$_{533-547}$ | No |
| #8 | Pre_2 | 6-1 | CASIRDRVADTQYF | 2-3 | 30 | CGTETTDSWGKLQF | 24 | RFNGIGVTQ NV$_{905-915}$ | DQA1*03:02-DQB1*03:03 | SEL1-like repeat protein [Bacteroidaceae bacterium] ‡ | LGVYYFNGI GVTQDQ$_{236-250}$ | No |
| #8 | Pre_3 | 27 | CATKGEANYGYTF | 1-2 | 12-3 | CAMSEMGTGFQKLVF | 8 | SIVRFPNI TNL$_{325-335}$ | DRA-DRB1*15:02 | LTA synthase family protein [Dechloromonas denitrificans] | LPGKSVVR WPNITNL$_{330-344}$ | No |
| #8 | Pre_5 | 5-1 | CASSLRTGELFF | 2-2 | 8-1 | CAVNGRNTGFQKLVF | 8 | NFTISVTTEI LPVSM$_{717-731}$ | DRA-DRB1*09:01 | Major capsid protein [Human papillomavirus 145] Periplasmic trehalase [Chlamydia bacterium] | NFTISVTTDA GDINE$_{350-364}$ LSTIVTTEIL PVD$_{288-301}$ | No |
| #8 | Pre_9 | 7-2 | CASAAGGTGGETQYF | 2-5 | 5 | CAETPFLSGTYKYIF | 40 | YIKWPWYIW LGFIAG$_{1209-1223}$ | DRA-DRB5*01:02 | Spike glycoprotein [Human coronavirus HKU1] | VKWPWYV WLLISFSF$_{1297-1311}$ | No |
| #8 | Pre_10 | 6-6 | CASSLGQGIHEQYF | 2-7 | 26-1 | CIVERGGSNYKLTF | 53 | SKRSFIEDL LFNKVT$_{813-827}$ | DPA1*01:03-DPB1*04:02 | Hypothetical protein, partial [Acinetobacter baumannii] Spike protein [Feline coronavirus] Spike protein [Canine coronavirus] | GKRSAVEDL LFNKVV$_{204-218}$ GKRSAVEDL LFNKVV$_{980-994}$ GKRSAVEDLL FNKVV$_{977-991}$ | No |
| #8 | Pre_14 | 4-3 | CASSQRQGAGDTQYF | 2-3 | 19 | CALSEAGIQGAQKLVF | 54 | IDRLITGRLQ SLQTY$_{993-1007}$ | DQA1*01:03-DQB1*06:01 | Excinuclease ABC subunit UvrA [Lentisphaeria bacterium] | VDRLITGRLE SSRLN$_{208-222}$ | No |
| #8 | Pre_15 | 20-1 | CSAKDRIYGYTF | 1-2 | 26-1 | CIVRSPSGSARQLTF | 22 | MIAQYTSAL LA$_{869-879}$ | DRA-DRB1*15:02 | MATE family efflux transporter [Selenomonas noxia] | ATIIAQYTSA LLALR$_{242-256}$ | No |
| #13 | Pre_5 | 4-3 | CASSQVSTGTGITGANVLTF | 2-6 | 5 | CARRSSSASKIIF | 3 | QNVLYENQ KLI$_{913-923}$ | DRA-DRB5*01:01 | Hypothetical protein [Neobacillus vireti] | TNVLYENQKL FLNLF$_{169-183}$ | No |
| #13 | Pre_8 | 18 | CASSPRAPPYEQYF | 2-7 | 21 | CAVRPAGGTGNQFYF | 49 | DKYFKNHTSP DVDLG$_{1153-1167}$ | DRA-DRB1*15:01 | Type VI secretion system contractile sheath large subunit [Salmonella enterica] | DYYFDHTSP DVDLLG$_{167-181}$ | No |
| #13 | Pre_12 | 4-2 | CASSQEGNTEAFF | 1-1 | 20 | CGCRGGTSYGKLTF | 52 | NVTWFHAIH VSGTNG$_{61-75}$ | DQA1*01:02-DQB1*06:02 | Dihydrofolate synthase [Actinobaculum sp. 313] | PQRSFHAIH VTGTNG$_{61-75}$ | No |
| #15 | Pre_1 | 20-1 | CSARDLTASAHGYTF | 1-2 | 17 | CATDAGQGGKLIF | 23 | SVTTEILP VSM$_{721-731}$ | DQA1*01:03-DQB1*06:01 | Hypothetical protein [Myxococcales bacterium] | PVTTEILPVSD DPPG$_{525-539}$ | No |
| #15 | Pre_2 | 24-1 | CATDSDLDQPQHF | 1-5 | 16 | CALSGYGSGYSTLTF | 11 | SVTTEILP VSM$_{721-731}$ | DQA1*01:03-DQB1*06:01 | Hypothetical protein [Myxococcales bacterium] | PVTTEILPVS DDPPG$_{525-539}$ | No |
| #15 | Pre_3 | 6-1 | CASDPKNGGEQYF | 2-7 | 29/DV5 | CAASVGFGNVLHC | 35 | FKIYSKH TPIN$_{201-211}$ | DRA-DRB5*01:02 | Uncharacterized protein APUU_31;289 S [Aspergillus puulaauensis] | CRAAFKLY SKHTPVE$_{123-137}$ | No |

*Table 6 continued on next page*

*Table 6 continued*

| Donor | Clonotype | TRBV | CDR3β | TRBJ | TRAV | CDR3α | TRAJ | S epitope | Restricting HLA | Cross-reactive antigen [species] | Cross-reactive peptide | Post-vaccinated expansion |
|---|---|---|---|---|---|---|---|---|---|---|---|---|
| #15 | Pre_4 | 19 | CASGLAGGNTGELFF | 2–2 | 10 | CVPSSGGYYNKLIF | 4 | QALNTLVK QLS957-967 | DRA-DRB1*08:02 | 4-hydroxybenzoate octaprenyltransferase [Pseudoduganella dura] | IQPLNTLVKQ LSVAA112-126 | No |
| #15 | Pre_5 | 6–5 | CASSAGLAGGGNTQYF | 2–3 | 5 | CAVISGSARQLTF | 22 | QALNTLV KQLS957-967 | DRA-DRB1*08:02 | 4-hydroxybenzoate octaprenyltransferase [Pseudoduganella dura] | IQPLNTLVKQ LSVAA112-126 | No |
| #15 | Pre_6 | 2 | CASVGGNEQFF | 2–1 | 9–2 | CALTRFVGGATNKLIF | 32 | RTFLLKYN ENGTITD273-287 | DRA-DRB1*15:02 | Unnamed protein product [Mytilus edulis] | NKKLLKYNE NGTFIT277-291 | No |
| #15 | Pre_7 | 4–1 | CASSHDGTPPDTQYF | 2–3 | 29/DV5 | CAAYSNYQLIW | 33 | FKIYSKHT PIN201-211 | DRA-DRB1*15:02 | Uncharacterized protein APUU_31,289 S [Aspergillus puulaauensis] | CRAAFKLYS KHTPVE123-137 | No |
| #15 | Pre_15 | 2 | CASSETGRGTDTQYF | 2–3 | 9–2 | CALYRGTYKYIF | 40 | LQSLQTYV TQQLIRA1001-1015 | DRA-DRB1*15:02 | Dyp-type peroxidase [Acinetobacter sp.] | CTVLQTYVTQ QLESV134-148 | No |
| #17 | Pre_7 | 6–1 | CASSLRGAFGYTF | 1–2 | 35 | CAGHLYGGSQGNLIF | 42 | NCTFEYVSQP FLMDL165-179 | DPA1*01:03-DPB1*04:02 | Fumarylacetoacetate hydrolase family protein [Alcaligenes faecalis] / Hypothetical protein [Planctomycetales bacterium] | ASLIEYVSQP FLLEP225-239 / AAGFEYVSQ PFSLPL533-547 | No |
| #17 | Pre_8 | 5–1 | CASSLNSGANVLTF | 2–6 | 13–1 | CAASIVQDQKLVF | 8 | LTPTWRVYS TGSNVF629-643 | DRA-DRB1*08:02 | Hypothetical protein [Novosphingobium chloroacetimidivorans] | APGTPTWRV YSTART277-291 | No |
| #17 | Pre_14 | 5–1 | CASSLGAGLYNEQFF | 2–1 | 38–1 | CAFINNNAGNMLTF | 39 | QALNTLVK QLS957-967 | DRA-DRB1*08:02 | 4-hydroxybenzoate octaprenyltransferase [Pseudoduganella dura] | IQPLNTLVKQ LSVAA112-126 | No |
| #17 | Pre_15 | 7–2 | CASSRTSGGTYEQYF | 2–7 | 25 | CAGQNTDKLIF | 34 | SIVRFPNI TNL325-335 | DRA-DRB1*15:01 | LTA synthase family protein [Dechloromonas denitrificans] | LPGKSWVR WPNITNL330-344 | Yes |

*Number ranges indicate the location of peptides in the proteins.

†Amino acids colored red indicate mismatches compared with corresponding S epitopes of Wuhan strain.

‡Antigen names and peptide sequences in cells with gray background indicate inactive antigens of the corresponding T clonotypes.

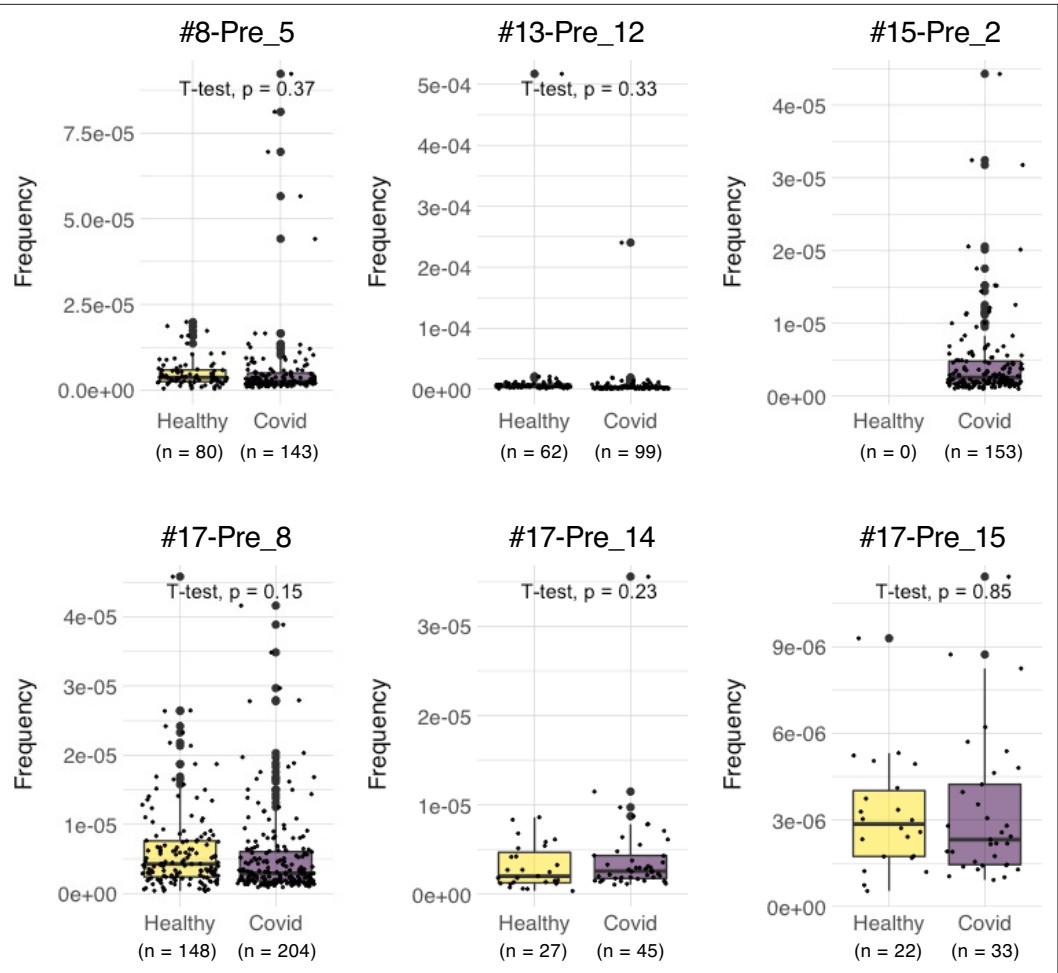

**Figure 6.** Frequencies of pre-existing S-reactive clonotypes in the public database of uninfected and infected cohorts. TCRβ sequences of the top expanded clonotypes in pre-vaccinated samples were investigated in the Adaptive database. Frequencies of detected clonotypes are shown in box plot. Healthy, dataset from 786 healthy donors. COVID, dataset from 1485 COVID-19 patients.

These observations suggest that these cross-reactive T cells might have been developed to establish tolerance against less harmful microbes, and thus unlikely to efficiently contribute to the protective viral immunity. Vaccination may induce opposite tendencies on T cell clonotypes that recognize the same antigen (*Aoki et al., 2022*), which is hardly detected by the bulk T cell analyses. The current study highlights the necessity of dynamic tracing of T cell responses in an epitope-specific clonotype resolution for the evaluation of vaccine-induced immunity.

The limitation of this study is the number of individuals we analyzed. However, chronological and clonological analysis of antigen-specific T cells in characteristic groups followed by epitope determination has not been performed before. This study suggests that mRNA vaccine is potent enough to prime rare T cell clonotypes that become dominant afterwards. Furthermore, we propose that the types of CD4[+] T clonotypes developed shortly after two doses of vaccination could be an indication of the longevity of antibodies in the following months. Tfh-inducing adjuvants or Tfh-skewing epitope would be a promising 'directional' booster in the post-vaccine era when most people worldwide were exposed to the same antigen in multiple doses within a short period. Furthermore, in addition to SARS-CoV-2, this strategy can also be applicable for the prevention of other infectious diseases of which neutralizing antibody titers are effective for protection.

## Materials and methods

**Key resources table**

| Reagent type (species) or resource | Designation | Source or reference | Identifiers | Additional information |
|---|---|---|---|---|
| Antibody | Anti-Human IgG HRP (Goat polyclonal) | Abcam | ab97175 | ELISA (1:5000) |
| Antibody | Anti-SARS-CoV-IgG WHO international Standard (Human polyclonal) | NIBSC | 20/136 | ELISA (10–31250) |
| Antibody | Anti-human CD3-FITC (mouse monoclonal) | BioLegend | Cat#: 300305 | FACS (1:100) |
| Antibody | TotalSeq-C anti-human Hashtags (mouse monoclonal, mixture) | BioLegend | Cat#: 394661, etc | Single-cell sequencing (1:50) |
| Antibody | Anti-mouse CD69-APC (armenian hamster monoclonal) | BioLegend | Cat#: 104513 | FACS (1:100) |
| Peptide, recombinant protein | SARS-CoV-2 Spike (trimeric) | Cell Signaling Technology | #65444 | |
| Peptide, recombinant protein | SARS-CoV-2 Nucleocapsid protein | ACRO Biosystems | NUN-C5227 | |
| Peptide, recombinant protein | SARS-CoV-2 (Spike Glycoprotein), PepMix | JPT Peptide Technologies GmbH | JER-PM-WCPV-S-1–2 | S peptide pool |
| Peptide, recombinant protein | Individual S peptide | Genscript | | a peptide scan (15mers with 11 aa overlap) through S protein (Swiss-Prot ID: P0DTC2) |
| Recombinant DNA reagent | pMX-IRES-rat CD2 (plasmid) | *Yamasaki et al., 2006* | | retroviral vector |
| Cell line (Cercopithecus aethiops) | Vero E6/TMPRSS2 | JCRB cell bank; *Yoshida et al., 2021* | | |
| Cell line (*Mus musculus*) | NFAT-GFP Reporter cell | *Matsumoto et al., 2021* | | T cell hybridoma lacking endogenous TCR with an NFAT-GFP reporter gene |
| Biological sample (Human gammaherpesvirus 4) | Epstein-Barr virus (EBV) | *Kanda et al., 2015* | | For B cell transformation |
| Software, algorithm | GraphPad Prism 8 | GraphPad Software | GraphPad Prism 8 | |

## Sample collection

Samples (serum, whole blood, and PBMCs) were collected four times at 0–7 days before 1st dose vaccination as pre-vaccination, at 14–21 days after 1st dose vaccination as 3 weeks sample, at 35–49 days after 1st dose vaccination as 6 weeks sample, and at 154–182 days after 1st dose of vaccination as 24 weeks sample. At the same time of blood sampling, adverse event information was also collected from all participants. PBMCs were isolated using BD vacutainer CPT cell separation tube (Beckton Dickinson), according to manufacturers' instructions. Isolated PBMCs were stored in the vapor phase of liquid nitrogen until use.

## Antibody titer determination by enzyme-linked immunosorbent assay (ELISA)

Serum antibody titer was measured using ELISA. Briefly, recombinant ancestral S protein (S1 + S2, Cell Signaling Technology; 1 µg/ml) or recombinant nucleocapsid protein (Acrobiosystems; 1 µg/ml) was coated on 96-well plate at 4 °C overnight. On the second day, wells were blocked with goat serum (Gibco) for 2 hr at room temperature. The sera were diluted from 10 to 31,250 folds in blocking buffer and incubated overnight at 4 °C. The next day, wells were washed and incubated with horseradish peroxidase (HRP)-conjugated antibodies (Abcam) for 3 hr at room temperature. After being washed with PBS-T (0.05% tween 20), wells were incubated with the peroxidase chromogenic substrate 3,3'–5,5'-tetramethyl benzidine (Sigma-Aldrich) for 30 min at room temperature, then the reaction was stopped by 0.5 N sulfuric acid (Sigma Aldrich). The absorbance of wells was immediately measured at 450 nm with a microplate reader (Bio-Rad). The value of the half-maximal antibody titer of each sample was calculated from the highest absorbance in the dilution range by using Prism 8

software. The calculated antibody titer was converted to BAU/ml by using WHO International Standard 20/136 (NIBSC) for ancestral S-specific antibody titer.

## Whole blood interferon-gamma release immune assay (IGRA) for SARS-CoV-2-specific T cell responses using QuantiFERON

SARS-CoV-2 specific T cell immune responses were evaluated by QuantiFERON SARS-CoV-2 (Qiagen) (*Jaganathan et al., 2021*), according to manufacturer's instructions, in which CD4[+] T cells were activated by epitopes coated on Ag1 tube, and CD4[+] and CD8[+] T cells were activated by epitopes coated on Ag2 tube. Briefly, 1 ml of whole blood sample with heparin is added into each of Nil (negative control), Mito (positive control), Ag1, and Ag2 tubes, and incubated at 37 °C for 22–24 hr. Tubes were then centrifuged at 3000×*g* for 15 min for collecting plasma samples. IFNγ derived from activated T cells was measured with enzyme-linked immunosorbent assay (ELISA) (Qiangen) according to the manufacturer's instructions. IFNγ concentration (IU/ml) was calculated with background (Nil tube) subtracted from values of Ag1 or Ag2 tubes.

## Pseudo-typed virus neutralization assay

The neutralizing activity of serum antibodies was analyzed with pseudo-typed VSVs as previously described (*Yoshida et al., 2021*). Briefly, Vero E6 cells stably expressing TMPRSS2 were seeded on 96-well plates and incubated at 37 °C for 24 h. Pseudoviruses were incubated with a series of dilutions of inactivated serum for 1 hr at 37 °C, then added to Vero E6 cells. At 24 hr after infection, cells were lysed with cell culture lysis reagent (Promega), and luciferase activity was measured by Centro XS[3] LB 960 (Berthold).

## In vitro stimulation of PBMCs

Cryopreserved PBMCs were thawed and washed with warm RPMI 1640 medium (Sigma) supplemented with 5% human AB serum (GeminiBio), Penicillin (Sigma), streptomycin (MP Biomedicals), and 2-mercaptoethanol (Nacalai Tesque). PBMCs were labeled with Cell Proliferation Kit (CellTrace Violet, ThermoFisher) following the manufacturer's protocol and were stimulated in the same medium with S peptide pool (1 µg/ml per peptide, JPT) for 10 days, with human recombinant IL-2 (1 ng/ml, Peprotech), IL-7 (5 ng/ml, BioLegend) and IL-15 (5 ng/ml, Peprotech) supplemented on day 2, day 5, and day 8 of the culture. On day 10 cells were washed and stained with anti-human CD3 and TotalSeq-C Hashtags antibodies. Proliferated T cells (CD3[+]CTV[low]) were sorted by cell sorter SH800S (SONY) and used for single-cell TCR and RNA sequencing analyses.

## Single-cell-based transcriptome and TCR repertoire analysis

Single cell library was prepared using the reagents from 10x Genomics following the manufacturer's instructions. After reverse transcription, cDNA was amplified for 14 cycles, and up to 50 ng of cDNA was used for construction of gene expression and TCR libraries. Libraries were sequenced in paired-end mode, and the raw reads were processed by Cell Ranger 6.0.0 (10x Genomics). Distribution of the mitochondrial gene percentage, n_counts and n_genes were fitted with a one-variable, two-component mixed Gaussian model using the Python package scikit-learn (*Pedregosa et al., 2011*) and divided into two distributions corresponding to high and low levels, respectively. The cutting threshold values were the middle value of the means of the two fitted Gaussian distributions. A package call Scrublet was also applied (*Wolock et al., 2019*), and the events whose main hashtag reads are less than 95% of the total hashtag reads were gated out before the UMAP plots were exported using BBrowser (*Le et al., 2020*). Tfh signature score was generated using canonical Tfh marker genes (*IL21, ICOS, CD200, PDCD1, POU2AF1, BTLA, CXCR5,* and *CXCL13*). Other cell populations were annotated using the following markers: Treg, *CD4[+]FOXP3[+]*; CD4T, *CD3E[+]CD4[+]*; CD8T, *CD3E[+]CD8A[+]*; central memory (cm) cells, *SELL*(CD62L)[hi] cells although sometimes *CCR7* expression is vague; effector memory (em) cells, *SELL[low/–]CCR7[–]* and *IFNG*-expressing cells containing populations; naïve cells, *CCR7[+]TCF7[+]*; cycling cells, *MKI67[hi]*; γδT, *TRDC[+]*; B cells, *CD19[+]*; Monocyte, *CD14[+]*; MAIT,

$CD3E^+KLRB1^+IL18R1^+$; Tfr, $FOXP3^+NRN1^+$ in cells with high Tfh score; CD4-CTL, $GZMB^+$ in CD4T cells (*Kaech et al., 2002*; *Meckiff et al., 2020*; *Sallusto et al., 2004*; *Wang et al., 2021*).

## Reporter cell establishment and stimulation

TCRα and β chain cDNA sequences were introduced into a mouse T cell hybridoma lacking TCR and having a nuclear factor of activated T-cells (NFAT)-green fluorescent protein (GFP) reporter gene (*Matsumoto et al., 2021*) using retroviral vectors (*Lu et al., 2021*; *Yamasaki et al., 2006*). TCR-reconstituted cells were co-cultured with 1 µg/ml of peptides in the presence of antigen-presenting cells (APCs). After 20 hr, cell activation was assessed by GFP and CD69 expression.

## Antigen-presenting cells

Transformed B cells and HLA-transfected HEK293T cells used as APCs were generated as described (*Lu et al., 2021*). For transformed B cells, $3×10^5$ PBMCs were incubated with the recombinant Epstein-Barr virus (EBV) suspension (*Kanda et al., 2015*) for 1 hr at 37 °C with mild shaking every 15 min. The infected cells were cultured in RPMI 1640 medium supplemented with 20% fetal bovine serum (FBS, CAPRICORN SCIENTIFIC GmbH) containing cyclosporine A (CsA, 0.1 µg/ml, Cayman Chemical). Immortalized B lymphoblastoid cell lines were obtained after 3 weeks of culture and used as APCs. For HLA-transfected HEK293T cells, plasmids encoding HLA class I/II alleles (*Jiang et al., 2013*) were transfected in HEK293T cells with PEI MAX (Polysciences).

## Determination of epitopes and restricting HLA

15-mer peptides with 11 amino acids overlap that cover the full length of S protein of SARS-CoV-2 were synthesized (GenScript). Peptides were dissolved in DMSO at 12 mg/ml and 12–15 peptides were mixed to create 26 different semi-pools. TCR-reconstituted reporter cells were stimulated with 1 µg/ml of S peptide pool (1 µg/ml per peptide, JPT), then 36-peptide pools that consist of three semi-pools each, then semi-pools, and then 12 individual peptides in the presence of autologous B cells to identify epitope peptides. To determine the restricting HLA, HLAs were narrowed down by co-culturing reporter cells with autologous and various heterologous B cells in the presence of 1 µg/ml of the epitope peptide. HLAs shared by activatable B cells were transduced in HEK239T cells and used for further co-culture to identify the restricting HLA.

## Statistics

All values with error bars are presented as the mean ± SEM. One-way ANOVA followed by Turkey's post hoc multiple comparison test was used to assess significant differences in each experiment using Prism 8 software (GraphPad Software). Differences were considered to be significant when p value was less than 0.05. p values in *Figure 6* were calculated with t-test using the 'stat_compare_means' function in R (version 4.3.0 for arm64).

## Acknowledgements

We thank S Iwai, Y Sakai, C Günther, J Sun, and Y Yanagida for experimental support, D Motooka, D Okuzaki, and YC Liu for bioinformatic data analysis and C Schutt and Y Yamagishi for discussion. This research was supported by Japan Agency for Medical Research and Development (JP223fa627002, JP223fa727001, JP23ym0126049 (SY), JP21ym0126049 (HN)) and Japan Society for the Promotion of Science Grants-in-Aid for Scientific Research (JP20H00505, JP22H05182, JP22H05183 (SY)). The Department of Health Development and Medicine is an endowed department supported by AnGes, Daicel, and FunPep.

## Additional information

### Competing interests

Hiroki Hayashi, Hironori Nakagami: Works in the Department of Health Development and Medicine, which is an endowed department supported by AnGes, Daicel, and FunPep. The other authors declare that no competing interests exist.

## Funding

| Funder | Grant reference number | Author |
|---|---|---|
| Japan Agency for Medical Research and Development | JP223fa627002 | Sho Yamasaki |
| Japan Agency for Medical Research and Development | JP21ym0126049 | Hironori Nakagami |
| Japan Society for the Promotion of Science | JP20H00505 | Sho Yamasaki |
| Japan Agency for Medical Research and Development | JP223fa727001 | Sho Yamasaki |
| Japan Agency for Medical Research and Development | JP23ym0126049 | Sho Yamasaki |
| Japan Society for the Promotion of Science | JP22H05182 | Sho Yamasaki |
| Japan Society for the Promotion of Science | JP22H05183 | Sho Yamasaki |

The funders had no role in study design, data collection and interpretation, or the decision to submit the work for publication.

## Author contributions

Xiuyuan Lu, Hiroki Hayashi, Eri Ishikawa, Formal analysis, Investigation, Visualization, Writing – original draft, Writing – review and editing; Yukiko Takeuchi, Julian Vincent Tabora Dychiao, Investigation; Hironori Nakagami, Conceptualization, Resources, Supervision, Funding acquisition, Writing – original draft, Project administration, Writing – review and editing; Sho Yamasaki, Conceptualization, Resources, Supervision, Funding acquisition, Methodology, Writing – original draft, Project administration, Writing – review and editing

## Author ORCIDs

Xiuyuan Lu ![ORCID] http://orcid.org/0000-0002-0784-2871
Hironori Nakagami ![ORCID] https://orcid.org/0000-0003-4494-3601
Sho Yamasaki ![ORCID] http://orcid.org/0000-0002-5184-6917

## Ethics

This project was approved by Osaka University Institutional Review Board (IRB) (reference No. 21487). 43 volunteers were enrolled in this project. Informed consent was obtained from all participants before the first blood sampling.

Reviewer #1 (Public Review): https://doi.org/10.7554/eLife.89999.4.sa1
Reviewer #3 (Public Review): https://doi.org/10.7554/eLife.89999.4.sa2
Author response https://doi.org/10.7554/eLife.89999.4.sa3

---

# Additional files

## Supplementary files
• MDAR checklist
• Source code 1. Scripts for analysis of single-cell sequencing data.

## Data availability

Single cell-based transcriptome data have been deposited in Gene Expression Omnibus (GEO) datasets (accession number GSE246535). Other data needed to support the conclusion of this manuscript are included in the main text and supplemental information.

The following dataset was generated:

| Author(s) | Year | Dataset title | Dataset URL | Database and Identifier |
|---|---|---|---|---|
| Lu X, Hayashi H, Motooka D, Nakagami H, Yamasaki S | 2023 | T cell responses in SARS-CoV-2 vaccinees with sustained or declined antibody titer | https://www.ncbi.nlm.nih.gov/geo/query/acc.cgi?acc=GSE246535 | NCBI Gene Expression Omnibus, GSE246535 |

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
