## [Editor Report · eLife assessment]

This **important** study by Lu et al aimed to determine the key factors of T cell responses associated with durable antibody responses following the initial two shots of COVID-19 mRNA vaccinations. By comparing the SARS-CoV-2 spike protein (S)-specific T cell subsets between ‘Ab sustainers’ and ‘Ab decliners’ that were present post-vaccination, the authors concluded that S-specific CD4^+^ T cells in ‘Ab sustainers’ were enriched with Tfh cells. There is **solid** evidence as the authors applied multiple methods and approaches to address the key questions, and the presented data are robust.

---

## [Referee Report · Reviewer #1 (Public Review)]

• A summary of what the authors were trying to achieve.

The authors cultured pre- and Post-vaccine PBMCs with overlapping peptides encoding S protein in the presence of IL-2, IL-7, and IL-15 for 10 days, and extensively analyzed the T cells expanded during the culture; by including scRNAseq, scTCRseq, and examination of reporter cell lines expressing the dominant TCRs. They were able to identify 78 S epitopes with HLA restrictions (by itself represents a major achievement) together with their subset, based on their transcriptional profiling. By comparing T cell clonotypes between pre- and post-vaccination samples, they showed that a majority of pre-existing S-reactive CD4+ T cell clones did not expand by vaccinations. Thus, the authors concluded that highly-responding S-reactive T cells were established by vaccination from rare clonotypes.

• An account of the major strengths and weaknesses of the methods and results.

Strengths

• Selection of 4 "Ab sustainers" and 4 "Ab decliners" from 43 subjects who received two shots of mRNA vaccinations.

• Identification of S epitopes of T cells together with their transcriptional profiling. This allowed the authors to compare the dominant subsets between sustainers and decliners.

Weaknesses were adequately addressed in the revised manuscript, and I do not have any additional concerns.

---

## [Referee Report · Reviewer #3 (Public Review)]

The paper aims to investigate the relationship between anti-S protein antibody titers with the phenotypes & clonotypes of S-protein-specific T cells in people who receive SARS-CoV2 mRNA vaccines. The paper recruited a cohort of COVID-19 naive individuals who received the SARS-CoV2 mRNA vaccines and collected sera and PBMCs samples on different time points. Then, three sets of data were generated: (1). Anti-S protein antibody titers on all time points. (2) Single-cell RNAseq/TCRseq analysis for divided T cells after in vitro stimulation by S-protein. (3) Peptide epitopes for each expanded TCR clone. Based on these, the paper reports two major findings: (A) Individuals having more sustained anti-S protein antibody response also have more Tfh-featured S-specific cells in their blood after 2nd-dose vaccination. (B). S-specific cross-reactive T cells exist in COVID-19 naive individuals, but most of these T cell clones are not expanded after SARS-CoV-2 vaccination.

The paper's strength is that it uses a very systemic strategy trying to dissect the relationship between antibody titers, T cell phenotypes, TCR clonotypes and corresponding epitopes. The conclusion is solid in general. However, the weaknesses include the relatively small sample size (4 sustainers vs. 4 decliners) and the use of in vitro stimulated cells for analysis, which may 'blur' the classification of T cell subsets. Nevertheless, it may have great impact on future vaccine design because it demonstrated that promoting Tfh differentiation is crucial for the longevity of antibody response. Additionally, this paper nicely showed that most cross-reactive clones that are specific to environmental/symbiotic microbes did not expand post- vaccination, providing important fundamental insights into the establishment of T-cell responses after SARS-CoV-2 vaccination.

---

## [Author Response]

The following is the authors’ response to the previous reviews.

**Reviewer #1 (Public Review):**
A summary of what the authors were trying to achieve.The authors cultured pre- and Post-vaccine PBMCs with overlapping peptides encoding S protein in the presence of IL-2, IL-7, and IL-15 for 10 days, and extensively analyzed the T cells expanded during the culture; by including scRNAseq, scTCRseq, and examination of reporter cell lines expressing the dominant TCRs. They were able to identify 78 S epitopes with HLA restrictions (by itself represents a major achievement) together with their subset, based on their transcriptional profiling. By comparing T cell clonotypes between pre- and post-vaccination samples, they showed that a majority of pre-existing S-reactive CD4+ T cell clones did not expand by vaccinations. Thus, the authors concluded that highly-responding S-reactive T cells were established by vaccination from rare clonotypes.An account of the major strengths and weaknesses of the methods and results.

Strengths:

Selection of 4 "Ab sustainers" and 4 "Ab decliners" from 43 subjects who received two shots of mRNA vaccinations.Identification of S epitopes of T cells together with their transcriptional profiling. This allowed the authors to compare the dominant subsets between sustainers and decliners.Weaknesses were properly addressed in the revised manuscript, and I do not have any additional concerns.

We appreciate the reviewer for the constructive comments and recommendations, which were a great help for us to improve our manuscript.

**Reviewer #3 (Public Review):**
Summary:The paper aims to investigate the relationship between anti-S protein antibody titers with the phenotypes&clonotypes of S-protein-specific T cells, in people who receive SARS-CoV2 mRNA vaccines. To do this, the paper recruited a cohort of Covid-19 naive individuals that receives the SARS-CoV2 mRNA vaccines and collect sera and PBMCs samples on different timepoints. Then they mainly generate three sets of data: (1). Anti-S protein antibody titers on all timepoints. (2) Single-cell RNAseq/TCRseq dataset for divided T cells after stimulation by Sprotein for 10 days. (3) Corresponding epitopes for each expanded TCR clones. After analyzing these result, the paper reports two major findings&claims: (A) Individuals having sustained anti-S protein antibody response also have more so-called Tfh cells in their single-cell dataset. (B). S-reactive T cells do exist before the vaccination, but they seems to be unable to response to Covid-19 vaccination properly.The paper's strength is it uses a very systemic and thorough strategy trying to dissect the relationship between antibody titers, T cell phenotypes, TCR clonotypes and corresponding epitopes, and indeed it reports several interesting findings about the relationship of Tfh clonotypes/sustained antibody and about the S-reactive clones that exist before the vaccination. The conclusion is solid in general but some claims are overstated. My suggestion is the authors should further limit their claims in abstract, for example,”Even before vaccination, S-reactive CD4+ T cell clonotypes did exist, most of which (MAY) cross-reacted with environmental or symbiotic bacteria" -- The paper don't have experimental evidence to show these TCR clones respond to these epitopes.

We thank the reviewer for pointing out the insufficient demonstration of experimental evidence. We have added the relevant data to Fig. S5 in the newly revised manuscript.

"These results suggest that de novo acquisition of memory Tfh-like cells upon vaccination (LIKELY) contributes to the longevity of anti-S antibody titers." --Given the small sample size and the statistical analysis was not significant, this claim was overstated."S-reactive T cell clonotypes detected immediately after 2nd vaccination polarized to follicular helper T (Tfh)-like cells (UNDER IN VITRO CULTURE)". -- the conclusion was based on vitro cultured cells, which had limitation.

We thank the reviewer for the helpful suggestion. We have corrected some sentences in line with these suggestions in the newly revised manuscript.

**Recommendations for the authors:**
Please note: Though most of the overstatement was removed from the original manuscript, authors still need to modify some of the statements in "Abstract".

We thank the reviewer for carefully reading our manuscript and giving us detailed suggestions. We have modified these statements in “Abstract” accordingly in the newly revised manuscript.